# *Alternaria alternata* JTF001 Metabolites Recruit Beneficial Microorganisms to Reduce the Parasitism of *Orobanche aegyptiaca* in Tomato

**DOI:** 10.3390/biology14020116

**Published:** 2025-01-23

**Authors:** Wenfang Luo, Xingxing Ping, Junhui Zhou, Shuaijun Gao, Xin Huang, Suqin Song, Jianjun Xu, Wei He

**Affiliations:** 1Xinjiang Key Laboratory of Agricultural Bio-Safety, Key Laboratory of Integrated Pest Management on Crops in Northwestern Oasis, Ministry of Agriculture and Rural Affairs, Institute of Plant Protection, Xinjiang Academy of Agricultural Sciences, Urumqi 830091, China; lf576263465@163.com (W.L.); junhuiqzhou@163.com (J.Z.); gaoshuaijun1998@163.com (S.G.); huangxin0924@126.com (X.H.); suqin_song@163.com (S.S.); 2Department of Microbiology, College of Life Sciences, Nankai University, Tianjin 300071, China; pxx13663441748@163.com

**Keywords:** disease suppression, *Orobanche aegyptiaca*, rhizosphere microbiome, tomato

## Abstract

The rhizosphere microbiome plays a crucial role in maintaining host health. Beneficial species in the rhizosphere can produce metabolites that directly suppress pathogens. However, the regulatory effects of these metabolites on the rhizosphere microbiome, and whether this regulation contributes to pathogen suppression, remain underexplored. In this study, we used high-throughput sequencing to investigate the differences in rhizosphere microbiomes between treatments with beneficial fungal metabolites and controls. Additionally, we employed isolation and in vitro inhibition assays to validate the role of beneficial fungal metabolites in recruiting beneficial microorganisms. Our findings provide valuable insights for the development of strategies that harness beneficial species’ metabolites to modulate rhizosphere microbiomes, potentially enhancing crop productivity.

## 1. Introduction

*Orobanche aegyptiaca* (*O. aegyptiaca)*, commonly known as broomrape, is an annual herbaceous plant in the Orobanchaceae family. Lacking leaves, chlorophyll, and functional roots, *O. aegyptiaca* is a holoparasite that completely relies on the root systems of its host plants for water, nutrients, and various growth hormones, thus posing a severe threat to host plant health [1]. In China, Xinjiang is the region most severely affected by *O. aegyptiaca*, with the parasite causing significant damage to crops such as processed tomatoes, melons, peppers, and cucurbits. The infestation spans an area of 53,000 to 67,000 hectares, resulting in annual losses exceeding 500 million RMB in agricultural production, and the affected area continues to expand [2,3]. In recent years, numerous biocontrol microorganisms have been found to play a critical role in helping plants resist diseases. Xi et al. demonstrated that *Pseudomonas mandelii* can inhibit the growth of *Orobanche cumana* [4]. *Pseudomonas fluorescens*, through the production of antibiotics and volatile organic compounds (VOCs), effectively controls a range of plant pathogens, nematodes, and insects [5]. VOCs produced by *Lysobacter* species have been shown to control the growth of *Phytophthora infestans* in vitro [6]. Additionally, *Bacillus* species contribute to biocontrol by suppressing plant pathogen growth, inducing systemic resistance in plants, and competing with pathogens for ecological niches [7].

The rhizosphere microbiome plays an essential role in plant health and disease resistance by facilitating nutrient acquisition [8], improving tolerance to abiotic stresses [9,10], and modulating immune responses to pathogens [11,12]. For example, arbuscular mycorrhizal fungi (AMF) enhance plant nutrient uptake and simultaneously bolster resistance against soil-borne diseases [13]. Moreover, both plant domestication [14,15,16] and breeding for disease resistance [17,18] influence the assembly of rhizosphere microbial communities, impacting their composition and functionality [19]. For instance, studies on the microbiomes of crop varieties with varying resistance to pathogens provide critical insights into the dynamic interplay between plants and their associated microbes [20,21]. Furthermore, pathogen inoculation has been shown to shape the root microbiome, altering its structure and function [22]. Microbial communities can alter host plant gene expression, enhance nutrient uptake, and activate defense pathways, thereby reducing parasite establishment and growth [23]. Advances in metagenomics and transcriptomics have revealed that specific microbial taxa and their functional genes are associated with induced systemic resistance (ISR) and pathogen suppression [21,24]. For example, genes involved in the synthesis of antimicrobial compounds, siderophores, and VOCs have been identified as key contributors to biocontrol efficacy [25,26,27].

Despite these advances, little is known about the impact of biocontrol agent metabolites on the composition of the rhizosphere microbiome. Specifically, it remains unclear whether and how these metabolites recruit beneficial microbes to establish disease-suppressive microbial communities. To test this hypothesis, we utilized the previously identified biocontrol fungus *Alternaria alternata* JTF001 (J1) against *O. aegyptiaca* [28] to conduct the following experiments: (i) evaluate the effects of J1 metabolite application on tomato seedling growth and the severity of *O. aegyptiaca* parasitism; (ii) investigate the role of the rhizosphere microbiome in mediating the suppression of *O. aegyptiaca* parasitism by J1 metabolites; (iii) assess the J1 metabolite-induced changes in the rhizosphere microbial community and isolated beneficial bacteria stimulated by J1 metabolites; and (iv) test the inhibitory effects of these beneficial bacteria on *O. aegyptiaca* germination in vitro.

## 2. Materials and Methods

### 2.1. Plants, Strain, Experimental Design, and Sample Collection

The seeds of *O. aegyptiaca* were collected from a single naturally matured *O. aegyptiaca* plant in a processing tomato field located in Jimusaer County (43°59′ N, 89°04′ E), Xinjiang, China. After collection, the seeds were air-dried naturally and stored at room temperature. The fungal strain *A. alternata* JTF001 was isolated and preserved by the Key Laboratory of Integrated Pest Management on Crops in Northwestern Oasis, Ministry of Agriculture, China. The strain was cultured in potato dextrose broth (PDB) at 28 °C and 180 rpm for five days to prepare the fermentation broth. A pot experiment was conducted to assess the inhibitory effect of the J1 fermentation broth on *O. aegyptiaca* parasitism. Three treatments were applied: J1 fermentation broth, a natural control (CK), and PDB (as a negative control) (Haibo Biotechnology Co., Ltd., Qingdao, Shangdong, China). Each treatment had six biological replicates. Tomato seedlings were transplanted into pots at the 5–6 leaf stage, with the growing medium consisting of a mixture of soil, vermiculite, and substrate (1:1:2). The seeds were surface-sterilized by immersion in 70% ethanol (Anjie Hi-Tech Disinfection Technology Co., Ltd., Dezhou, Shangdong, China) for 1 min, followed by 2.5% sodium hypochlorite (Wanjia Standard Material R&D Center Co., Ltd., Xinxiang, Henan, China) for 3 min, and thoroughly rinsed with sterile water. Before transplanting tomato seedlings, approximately 25 mg (around 5000 seeds) of sterilized *O. aegyptiaca* seeds were evenly mixed with a small amount of soil and spread across the surface of the pots. Two days after transplanting, the seedlings were treated with 70 mL of fermentation broth (at a concentration of approximately 5 × 10^5^ cfu/mL) per pot. This application was repeated every 7 days, for a total of four treatments. The potted plants were placed in a solar greenhouse, with an average daytime temperature of 28–30 °C and an average nighttime temperature of 23–25 °C. During this period, no water was applied; only the fermentation broth was irrigated. Forty days post-transplantation, the tomato plants were harvested, and plant height as well as *O. aegyptiaca* infestation levels were recorded. To obtain the rhizosphere soil, we carefully uprooted the plants and gently tapped them to remove soil adhering to the roots, following the method described by Wei et al. [29]. The infestation levels of O. aegyptiaca were recorded quantitatively by counting the number of shoots per plant [30].

### 2.2. DNA Extraction, Amplicon Sequencing, and Data Preprocessing

DNA was extracted using the TGuide S96 Magnetic Soil/Stool DNA Kit (Tiangen Biotech, Beijing, China) following the manufacturer’s protocol. DNA concentrations were measured with the Qubit dsDNA HS Assay Kit and the Qubit 4.0 Fluorometer (Invitrogen, Thermo Fisher Scientific, Portland, OR, USA).

To assess the diversity of soil bacteria and fungi, 16S rRNA gene and nuclear ribosomal ITS amplicon sequencing was performed using the PacBio Sequel II platform. For bacterial community analysis, full-length 16S rRNA genes were amplified using the universal primer set 27F (AGRGTTTGATYNTGGCTCAG) and 1492R (TASGGHTACCTTGTTASGACTT). For fungal community analysis, the full-length internal transcribed spacer (ITS) regions were amplified with the primers ITS1F (5′-CTTGGTCATTTAGAGGAAGTAA-3′) and ITS4 (5′-TCCTCCGCTTATTGATATGC-3′). The amplification protocols differed for bacterial 16S and fungal ITS regions. For full-length bacterial 16S amplification, the protocol included an initial denaturation at 95 °C for 2 min, followed by 25 cycles of denaturation at 98 °C for 10 s, annealing at 55 °C for 30 s, and extension at 72 °C for 1 min 30 s, with a final extension at 72 °C for 2 min. For full-length fungal ITS amplification, the protocol started with an initial denaturation at 95 °C for 5 min, followed by 8 cycles of denaturation at 95 °C for 30 s, annealing at 55 °C for 30 s, and extension at 72 °C for 45 s. Subsequently, an additional 24 cycles were performed with denaturation at 95 °C for 30 s, annealing at 60 °C for 30 s, and extension at 72 °C for 45 s, concluding with a final extension at 72 °C for 5 min. PCR amplicons were purified using Agencourt AMPure XP Beads (Beckman Coulter, Indianapolis, IN, USA) and quantified with the Qubit dsDNA HS Assay Kit and Qubit 4.0 Fluorometer (Invitrogen, Thermo Fisher Scientific). After quantification, amplicons were pooled in equimolar amounts. SMRTbell libraries were prepared using the SMRTbell Express Template Prep Kit 2.0 (Pacific Biosciences, Menlo Park, CA, USA) according to the manufacturer’s instructions. Purified SMRTbell libraries from pooled and barcoded samples were sequenced on a PacBio Sequel II 8M cell using the Sequel II Sequencing Kit 2.0.

Our data processing approach was adapted from the methods described in Ping et al. [21], with some modifications. Primer sequences were removed, and sequence quality filtering was performed using VSEARCH (--fastx_filter, --fastq_maxee 0.5) [31]. Biological reads were identified at 100% sequence similarity using the unoise3 [32] algorithm under default settings. Each zero-radius operational taxonomic unit (zOTU) was assigned a taxonomic classification using the SINTAX algorithm (-sintax with -strand both and -sintax_cutoff 0.8, usearch) in USEARCH, against the Silva v.138 (for bacteria) and UNITE v.8.3 (for fungi) databases [33,34]. zOTUs assigned to mitochondria, cyanobacteria, and chloroplasts were removed from the dataset. The remaining sequences were used to construct a zOTU table (-otutab, -zotus, usearch). For downstream analysis, sequence counts were rarefied to a uniform depth of 2000 reads per sample for both 16S rRNA gene and ITS sequences using QIIME v1.91 (single_rarefaction.py) [35].

### 2.3. Isolation and Identification of Culturable Bacteria, and In Vitro Inhibition Assays

Rhizosphere soil from tomato plants treated with J1 metabolites was serially diluted, and aliquots of the dilutions were spread onto NA medium. Bacteria were isolated from soil suspension concentrations of 10^4^, 10^5^, and 10^6^. After incubation at 28 °C for 24 h, single colonies were selected for further analysis. These isolates were taxonomically classified using 16S rRNA gene sequencing with primers 27F/1492R [36]. The resulting 16S rRNA gene sequences were aligned using NCBI’s Nucleotide BLAST tool (https://www.ncbi.nlm.nih.gov/, accessed on 18 December 2024) to determine the approximate phylogenetic affiliations of the strains. To evaluate the inhibitory effect of the isolated strain on *O. aegyptiaca* seed germination, we conducted a microcosm experiment. Briefly, tomato seedlings at the 5–6 true leaf stage were carefully removed from sterilized soil, immersed in 0.5% sodium hypochlorite for 20 s for surface sterilization, and rinsed thoroughly with sterile water. The seedlings were then placed on sterile petri dishes lined with autoclaved cotton. Approximately 15 mg of surface-sterilized *O. aegyptiaca* seeds were evenly distributed around the roots. Subsequently, 3 mL of the isolated strain suspension (1 × 10^7^ CFU/mL) was applied to the area surrounding the *O. aegyptiaca* seeds. Sterile water was used as a control. The germination status of *O. aegyptiaca* seeds was assessed after 7 days.

### 2.4. Statistical Analysis

To investigate differences in microbial communities, principal coordinate analysis (PCoA) based on the Bray–Curtis dissimilarity index was conducted to assess taxonomic dissimilarities among treatments. Statistical significance of these dissimilarities was tested using PERMANOVA. Network analysis was performed and visualized using the igraph package [37] and ggplot2 [38]. A linear modeling approach was applied to evaluate variations in microbial taxa abundance between J1 and CK treatments. Descriptive statistics were generated using the R statistical environment (v4.3.3; http://www.r-project.org/, accessed on 18 December 2024) and GraphPad Prism software (v8.0).

### 2.5. Data Availability

The sequences for 16S rRNA gene and ITS have been deposited in the National Center for Biotechnology Information (NCBI; https://www.ncbi.nlm.nih.gov/, accessed on 18 December 2024) under the accession number PRJNA1182506 and PRJNA1182524, respectively.

## 3. Results

### 3.1. J1 Metabolites Impact on O. aegyptiaca Parasitism and Plant Growth

The application of J1 metabolites demonstrated a pronounced inhibitory effect on *O. aegyptiaca* populations compared to the control treatment without J1 metabolites and the treatment with PDB medium (Figure 1). Specifically, the introduction of J1 metabolites significantly reduced the number of *O. aegyptiaca* individuals (Figure 1A). Although the PDB treatment also resulted in a reduction of *O. aegyptiaca* numbers relative to the control, this decrease was not statistically significant. Furthermore, the application of J1 metabolites notably enhanced plant growth compared to both the control and PDB treatments, while no significant differences were observed between the control and PDB treatments (Figure 1B). These findings suggest that J1 metabolites not only suppress *O. aegyptiaca* parasitism but also promote the growth of host plants.

### 3.2. J1 Metabolites Reshape the Structure and Composition of the Tomato Rhizosphere Microbiome

To investigate the impact of J1 metabolites on the rhizosphere microbiome of tomato, we conducted an unconstrained principal coordinate analysis, which revealed a distinct response of the bacterial community to the J1 metabolites (Figure 2). Compared to the control treatment (CK), the application of J1 metabolites significantly altered the structure of the rhizosphere bacterial community (*p* < 0.05), as indicated by 16S rRNA sequencing, whereas no significant effects were observed on the structure and composition of the fungal community (Figure 2A,B and Appendix A).

Subsequently, we analyzed the effects of J1 metabolites on the composition of the rhizosphere microbiome at the phylum level (Figure 2). For bacteria, the results indicated an increase in the abundance of Proteobacteria and Firmicutes compared to CK. These phyla also exhibited increased abundance relative to the PDB treatment. Conversely, the abundance of Actinobacteriota, Bacteroidota, Acidobacteriota, Myxococcota, Gemmatimonadota, Patescibacteria, and Chloroflexi was found to decrease in the J1-treated rhizosphere (Figure 2C). Regarding fungi, the abundance of Ascomycota and Basidiomycota decreased following the application of J1 metabolites compared to CK, with Ascomycota levels also being lower than those in the PDB treatment. Notably, the J1-treated rhizosphere exhibited increased abundance of Rozellomycota, Mortierellomycota, and Chytridiomycota (Figure 2D). In summary, our findings demonstrate that the application of J1 metabolites significantly alter the structure and composition of the tomato rhizosphere bacterial community, while the effects on the fungal community structure and composition are not significant. These results suggest that the inhibitory effect of J1 metabolites on *O. aegyptiaca* may be mediated through the modulation of the tomato rhizosphere bacterial community.

### 3.3. J1 Metabolites Restructure Microbial Co-Occurrence Networks

To compare the effects of J1 metabolite application on the interaction patterns of the tomato rhizosphere microbiome, we constructed the inter-kingdom co-occurrence network of bacteria and fungi using the igraph package. The results indicated that the application of J1 metabolites significantly altered the structure of the co-occurrence network, enhancing its complexity (Figure 3). Specifically, the J1-treated network exhibited the highest complexity, comprising 198 nodes and 185 edges, with an average degree of 5.298 (Figure 3A and Appendix A). Additionally, we assessed the stability of the networks by randomly removing nodes, revealing that the J1-treated soil network exhibited greater stability than the CK soil network, as determined by both average degree and natural connectivity measures (Figure 3 and Appendix A). Network analysis results indicate that the application of J1 metabolites enhances the stability of the tomato rhizosphere microbial interaction network, thereby providing a more favorable microbial environment for plant growth.

### 3.4. Identification of Candidate Beneficial Taxa Associated with Plant Growth Promotion

To identify beneficial taxa associated with the inhibition of *O. aegyptiaca* growth and the promotion of plant growth, we conducted an abundance differential analysis at the zOTU level (Figure 4A,B). Our analysis revealed an enrichment of zOTUs belonging to the genera *Sphingomonas*, *Rhodanobacter*, *Reyranella*, *Pseudomonas*, *Pseudarthrobacter*, *Pedomicrobium*, *Niastella*, *MND1*, *Lysobacter*, *Luteimonas*, *Haliangium*, *Dongia*, *Devosia*, *Burkholderia−Caballeronia−Paraburkholderia*, *Bdellovibrio*, *Bauldia*, *Bacillus*, *Arthrobacter*, and *Allorhizobium−Neorhizobium−Pararhizobium−Rhizobium* in the J1 treatment (Figure 4D). Notably, we found that zOTUs of the genera *Pseudomonas* and *Bdellovibrio* were exclusively enriched in the J1 treatment. For fungi, we observed an enrichment of fungal zOTUs belonging to *Neobulgaria*, *Lasiobolidium*, *Humicola*, *Heydenia*, *Gibberella*, *Fusarium*, *Cutaneotrichosporon*, *Cladosporium*, *Chrysosporium*, *Chaetomium*, and *Alternaria* in the J1 treatment (Figure 4E). To further refine our candidate taxa, we integrated the results from the J1 microbial co-occurrence network and selected zOTUs ranked in the top 6.6% based on degree as key taxa within the network. By combining the results of the abundance differential analysis and the key network taxa, we ultimately identified three potential candidates: zOTU_388, zOTU_533, and zOTU_2335 (Figure 4C and Appendix A). Interestingly, all three belong to the genus *Pseudomonas*.

We sought to isolate these species and evaluate their inhibitory effects on *O. aegyptiaca*. Among the 11 isolated *Pseudomonas* strains, we selected three candidate strains due to their high sequence similarity with zOTU_388 (FJT-54: 97.78%), zOTU_533 (FJT-10: 96.67%), and zOTU_2335 (FJT-49: 97.56%). Our findings revealed that all three strains exhibited significant inhibitory effects on the germination of *O. aegyptiaca*, resulting in abnormal germ tube development (Figure 5). In the treatments with *Pseudomonas* strains, the germ tubes of *O. aegyptiaca* displayed growth impairment, accompanied by wilting and changes in color (Figure 5B–D). These results demonstrate that J1 metabolites promote plant health by recruiting beneficial *Pseudomonas* strains that suppress the growth of *O. aegyptiaca*.

## 4. Discussion

The results of this study demonstrate that J1 metabolites exert a significant effect on both inhibiting *O. aegyptiaca* growth and promoting host plant growth. Specifically, the application of J1 metabolites led to a marked reduction in the number of *O. aegyptiaca*, a suppression not observed in the PDB medium-treated or CK groups. This inhibitory effect of J1 metabolites on *O. aegyptiaca* is consistent with earlier studies, which suggest that microbial metabolites can enhance host plant vitality by interfering with the germination or attachment of parasitic plants [4,39,40].

To further explore the impact of J1 metabolites on rhizosphere microbial communities, 16S rRNA sequencing analysis revealed that the application of J1 metabolites significantly altered the structure of the rhizosphere bacterial community. Notably, the abundance of Proteobacteria and Firmicutes was significantly increased—taxa typically associated with plant growth promotion and disease suppression [41,42,43]. This suggests that J1 metabolites may enhance plant health by optimizing the microbial environment. Plants often regulate the composition of their root microbiomes to cope with environmental stress. For example, continuous monocropping of wheat or inoculation with *Rhizoctonia solani* has been shown to enrich beneficial bacterial genera such as *Chitinophaga*, *Pseudomonas*, *Chryseobacterium*, and *Flavobacterium* [44]. Liu et al. demonstrated that susceptible banana varieties can suppress pathogens by recruiting members of *Trichoderma* and *Penicillium* [45]. In contrast, we observed a decrease in the abundance of taxa such as Actinobacteriota and Bacteroidota, indicating that J1 metabolites may selectively enrich beneficial microbes while suppressing the expansion of other microbial communities. This selective microbial modulation is consistent with previous studies, highlighting the ability of plants to selectively enrich beneficial microbes through metabolic compounds, a process that greatly promotes plant health [46,47,48].

The fungal community analysis indicates that while J1 metabolites did not significantly affect the overall structure of the fungal community, their regulation of fungal abundance is noteworthy. Following the application of J1 metabolites, the abundance of Ascomycota and Basidiomycota decreased, suggesting that these taxa may harbor potential pathogens, and their reduction likely contributes to enhanced plant health. *Rhizoctonia solani*, a common pathogen within Ascomycota, and *Botrytis cinerea*, which causes gray mold and poses a serious threat to various crops and horticultural plants globally, are both members of these fungal groups [49,50]. The observed decrease in the abundance of Ascomycota and Basidiomycota may reduce the likelihood of co-infection by these pathogens, thereby mitigating their pathogenicity [51]. In contrast, the increase in Rozellomycota and Mortierellomycota, although their precise functions are not fully understood, may be associated with beneficial processes such as organic matter decomposition [52,53].

Inter-kingdom interaction network analysis provided further insights into the complex relationships among microbes. We found that J1 metabolites not only altered the composition of the rhizosphere microbiome but also increased the complexity and stability of the microbial community. Specifically, after J1 application, both the number of nodes and edges in the interaction network increased, with network stability significantly higher than that of the control group. The enhancement of network stability may result from the direct or indirect influence of J1 metabolites on the recruitment of beneficial microbes to the rhizosphere, thereby strengthening interactions within the microbial community [54]. This enhanced complexity and stability likely offer plants a more stable microbial environment, supporting growth and adaptation under stress. Ping et al. demonstrated that the complexity and stability of the rhizosphere microbiome co-occurrence network in resistant cabbage varieties were higher than in susceptible varieties [21]. However, when microbial network stability is disrupted, the root environment becomes more susceptible to pathogen invasion [55]. Notably, through differential abundance analysis and co-occurrence network screening, we identified several potential beneficial microbial taxa. Among these, *Pseudomonas* and *Bdellovibrio* are linked to plant growth promotion and disease suppression [56,57,58]. Further network analysis revealed that the top 6.6% of taxa, based on degree in the J1 network, were key contributors. Among these, three of the key zOTUs belonged to the *Pseudomonas* genus. Members of the genus *Pseudomonas* are widely recognized as effective biocontrol agents for suppressing soil-borne plant diseases [59,60]. Aligned with these findings, our in vitro inhibition assays revealed that all three strains significantly suppressed the germination of *O. aegyptiaca*. These results underscore the pivotal role of *Pseudomonas* in suppressing *O. aegyptiaca* parasitism and promoting plant growth through the action of J1 metabolites. Plant root exudates, as vital sources of nutrients and signaling molecules for rhizosphere microbial communities, are crucial for the recruitment of beneficial microbes. Different plant genotypes often exhibit distinct metabolic profiles, which in turn shape the composition of their root-associated microbiomes. Valerie et al. demonstrated that different sorghum genotypes produce varying root exudates, thereby influencing the structure of their rhizosphere microbiota [61]. In the poplar rhizosphere, microbial colonization alters the primary metabolites of the root system, further modulating the abundance and composition of the rhizosphere microbiome [62]. Additionally, specialized triterpenoid metabolites shape the species-specific microbial community composition in *Arabidopsis* [63]. Plants can stimulate the microbial communities that degrade organic matter by secreting specific compounds, thereby enhancing their activity, improving nutrient acquisition, and promoting plant growth [64,65,66]. However, the specific compounds within tomato root exudates that are stimulated by J1 metabolites to recruit *Pseudomonas* and reshape the rhizosphere microbiome remain unknown. Future investigations into the composition of tomato root exudates could provide valuable insights into the mechanisms by which plants contribute to the establishment of J1 metabolite-induced soil suppressiveness.

## 5. Conclusions

This study highlights that J1 metabolites play a pivotal role in reshaping the tomato rhizosphere microbiome, fostering the recruitment of beneficial microbes that enhance resistance to *O. aegyptiaca* parasitism. These findings provide a foundation for leveraging J1 metabolites as a novel biocontrol strategy, offering new avenues for the development of sustainable agricultural practices aimed at mitigating parasitic plant infestations. Future studies should focus on characterizing the composition of tomato root exudates to gain a deeper understanding of how plants contribute to the establishment of J1 metabolite-induced soil suppressiveness. Additionally, exploring the interactions between J1 metabolites and tomato root exudates, as well as their role in modulating the recruitment of beneficial microbes, will provide valuable insights into the underlying mechanisms. Investigating the genetic pathways involved in plant–microbe signaling could further elucidate how J1 metabolites influence microbial community dynamics. Finally, expanding these studies to encompass other crop species and varying environmental conditions will help assess the broader applicability of J1 metabolite-induced soil suppressiveness for sustainable agricultural practices.

## Figures and Tables

**Figure 1 biology-14-00116-f001:**
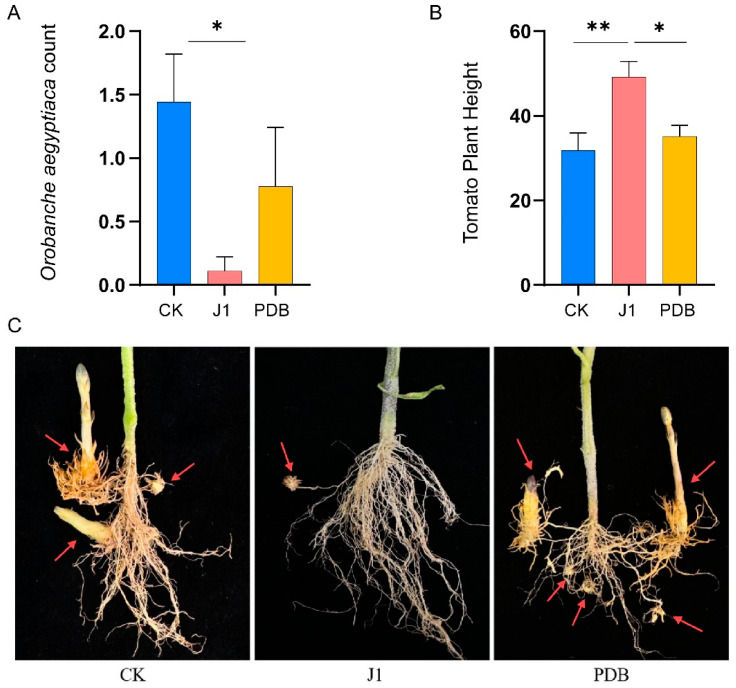
Inhibition of *Orobanche aegyptiaca* parasitism by J1 metabolites. (**A**) Effects of different treatments on *O. aegyptiaca* parasitism. (**B**) Plant height of tomato under different treatments; *, *p* < 0.05; **, *p* < 0.01. (**C**) Impact of different treatments on *O. aegyptiaca* parasitism in tomato;The arrow points to the parasitic *Orobanche aegyptiaca*. J1: *Alternaria alternata* JTF001 metabolite treatment; PDB: potato dextrose broth treatment; CK: control treatment.

**Figure 2 biology-14-00116-f002:**
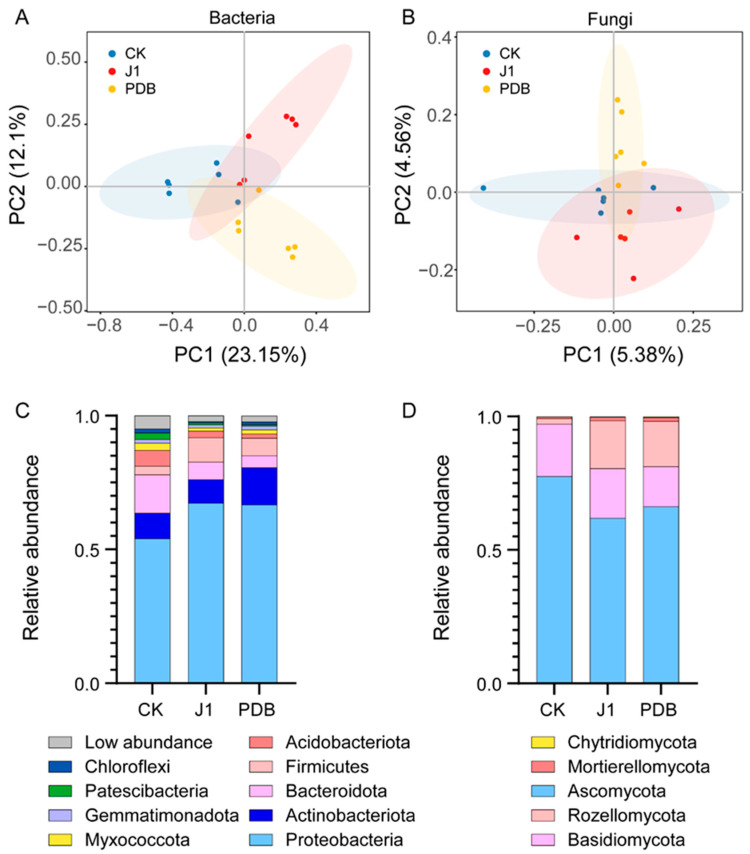
Differences in tomato rhizosphere microbiota under *Orobanche aegyptiaca* parasitism across different treatments. Principal coordinate analysis (PCoA) showing differences in microbial community structure based on Bray–Curtis distances among different treatments: (**A**) bacterial communities; (**B**) fungal communities. Taxonomic composition of bacterial (**C**) and fungal (**D**) communities under different treatment conditions. J1: *Alternaria alternata* JTF001 metabolite treatment; PDB: potato dextrose broth treatment; CK: control treatment.

**Figure 3 biology-14-00116-f003:**
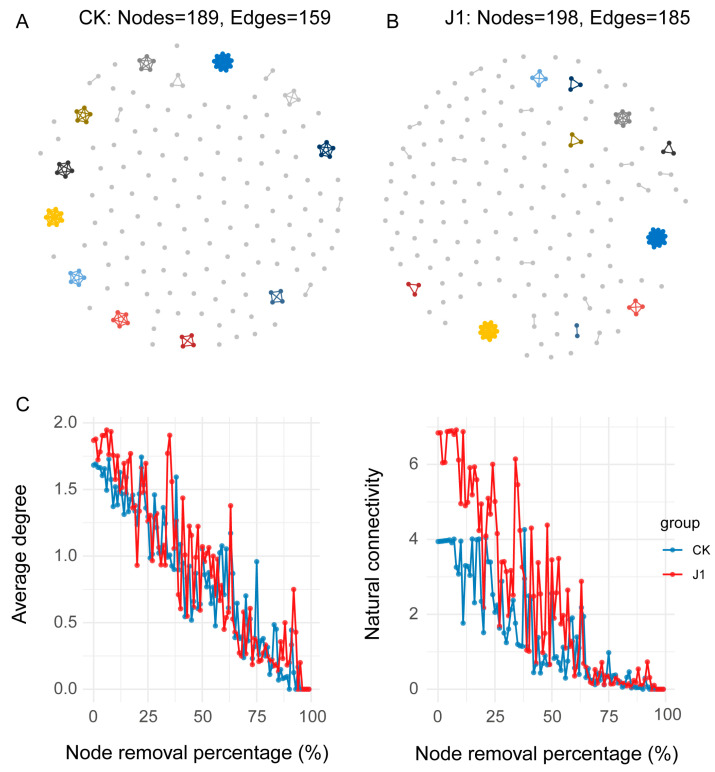
Microbial co-occurrence networks in the J1 and the CK group. An overview of the CK (**A**) and J1 (**B**) co-occurrence networks. Each node represents a unique microbial zOTUs. Each connection between the 2 nodes represents a strong cooccurrence relationship (Spearman’s r > 0.3 and *p* < 0.05). Different colors indicate different modules. (**C**) The robustness of the ecological networks in the J1 and the CK treatment based on node removal to simulate species extinction. J1: *Alternaria alternata* JTF001 metabolite treatment; CK: control treatment.

**Figure 4 biology-14-00116-f004:**
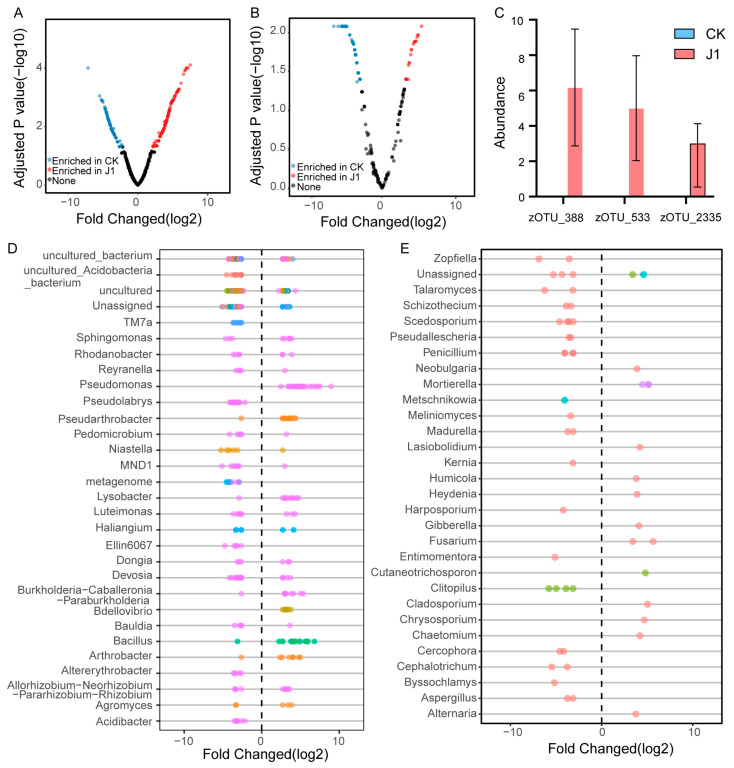
Analysis of abundance differences between different plant treatments. Enrichment (positive) and depletion (negative) of bacterial (**A**) and fungal (**B**) zOTUs across different plant treatments. Each point represents an individual zOTU. (**C**) Abundance information of candidate strains under different plant treatments. (**D**) Top 30 genera of enriched or depleted bacterial (**D**) and fungal (**E**) zOTUs across different plant treatments. Point colors represent phylum classification. J1: *Alternaria alternata* JTF001 metabolite treatment; CK: control treatment.

**Figure 5 biology-14-00116-f005:**
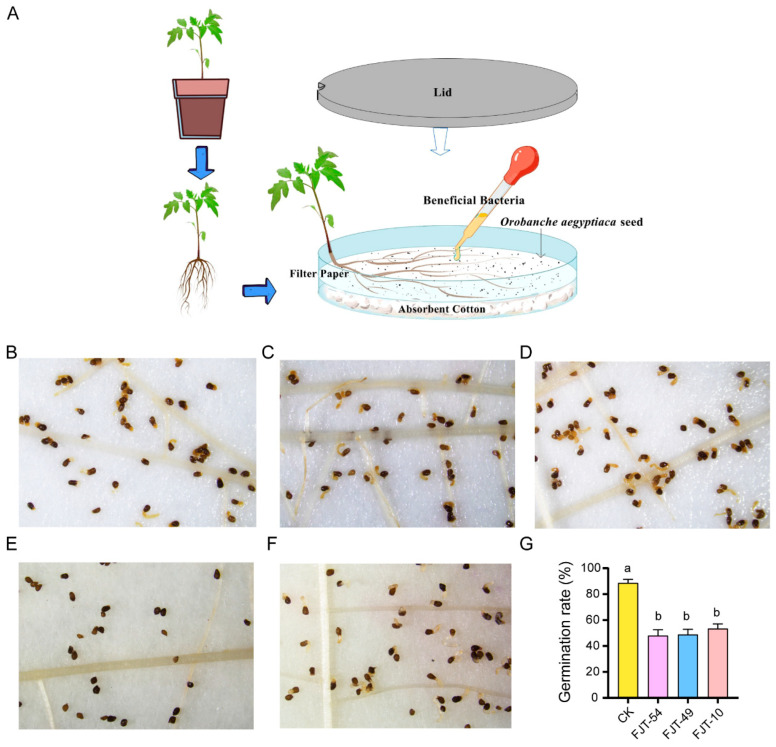
Effects of bacterial inoculation on *Orobanche aegyptiaca* germination. (**A**) Schematic representation of the microcosm experimental model. Germination rate of *O. aegyptiaca* after adding bacterial strains (FJT-54 (**B**), FJT-49 (**C**), FJT-10 (**D**), and J1 (**E**)) and water (H_2_O, negative control, **F**). (**G**) Comprehensive statistics of all processed germination rates; a, b: indicate significant differences. J1: *Alternaria alternata* JTF001 metabolite treatment; CK: control treatment.

## Data Availability

The sequences for 16S rRNA gene and ITS have been deposited in the National Center for Biotechnology Information (NCBI; https://www.ncbi.nlm.nih.gov/, accessed on 18 December 2024) under the accession number PRJNA1182506 and PRJNA1182524, respectively.

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
