# Peer review of "Alternaria alternata JTF001 Metabolites Recruit Beneficial Microorganisms to Reduce the Parasitism of Orobanche aegyptiaca in Tomato"

_biology, 2025, doi:10.3390/biology14020116_

Round 1
Reviewer 1 Report
Comments and Suggestions for Authors
I had the privilege of reviewing the aforementioned manuscript, which presents an interesting and innovative study. While the manuscript has significant merit, several revisions are necessary prior to its publication to enhance clarity and alignment with scientific standards.
The abstract requires reorganization for better clarity and conciseness. Incorporating numerical data to highlight key findings is recommended.
The keywords should be presented in alphabetical order, adhering to standard formatting practices.
Fig. 1C: Images of tomato roots are included, but the corresponding textual explanation is insufficient. Additional details on the significance of these images should be provided.
Fig. 3A: A network diagram is presented; however, it lacks sufficient detail. The discussion should explore how network stability could be enhanced through the application of J1 metabolites.
Fig. 5B-F: Images depicting plant seed tube development are included but inadequately described. Distinctions between normal and abnormal tube development should be clearly marked and elaborated upon.
While the extensive use of figures improves comprehension, detailed textual explanations accompanying each figure are necessary for better interpretability.
The discussion is comprehensive and well-aligned with the results. However, the conclusion section could benefit from the inclusion of statements outlining future research perspectives.
Minor typographical errors were observed throughout the manuscript. These should be addressed during the revision process.
In-text citations do not consistently adhere to the journal's formatting guidelines. This should be rectified.
Scientific names must be italicized where applicable (e.g., Lines 218, 421, 426).
The formatting of article titles in the reference list is inconsistent. Some titles begin with uppercase letters, while others start with lowercase letters. All references should follow a uniform format as per the journal’s guidelines (e.g., Lines 362, 385, 407).
In summary, the manuscript demonstrates significant scientific value. Addressing the outlined points will substantially improve its quality and presentation.
Author Response
Thank you for taking the time to thoroughly review our manuscript and provide valuable insights. We sincerely appreciate your efforts to help us enhance the clarity and robustness of our work. In response to your detailed comments and suggestions, we have implemented revisions using the track changes mode to highlight these modifications and improve the manuscript's clarity and overall quality. We believe these revisions have significantly enhanced the presentation and comprehensibility of our research. Below, we address each of your comments in detail:
Comment 1: The abstract needs reorganization to make it clearer and more concise. Consider incorporating numerical data to highlight key findings.
Response 1: Thank you for your valuable suggestion. We have reorganized the abstract to improve clarity and conciseness, ensuring a more logical flow of information. Additionally, we have included numerical data to emphasize key findings and enhance the presentation of results. We believe these revisions have improved the overall quality and readability of the abstract (Ln 12–28).
Comment 2: Keywords should be listed in alphabetical order, following standard formatting conventions.
Response 2: Thank you for your valuable suggestion. The keywords have been revised accordingly.
Comment 3: Figure 1C: Includes images of tomato roots, but the accompanying text lacks sufficient explanation. More details about the significance of these images should be provided.
Response 3: We sincerely appreciate your valuable feedback. We have revised the legend for Figure 1C to provide more details about the significance of the tomato root images (Ln 196). The updated description now includes a clearer explanation of how these images contribute to the overall findings and their relevance to the study. We hope this improves the clarity and comprehensiveness of the figure presentation.
Comment 4: Figure 3A: A network diagram is provided, but it lacks sufficient detail. The discussion should explore how the application of the J1 metabolite enhances network stability.
Response 4: Thank you for your insightful comment. We have revised Figure 3A to include more details, making the network diagram clearer and more informative (Ln 246). In the discussion section, we have expanded on the potential mechanisms by which the application of the J1 metabolite may enhance network stability (Ln 340–343).
Comment 5: Figures 5B–F: These include images depicting plant seed tube development, but the description is insufficient. Clear labeling and detailed explanations distinguishing normal and abnormal tube development are needed.
Response 5: Thank you for your observation. We have made the suggested modifications (Ln 285–287 and Ln 291–295).
Comment 6: Ln 47. While extensive use of figures improves comprehension, each figure requires detailed textual explanations for better interpretability.
Response 6: Thank you for your valuable suggestion. We have incorporated detailed textual explanations for each figure in the manuscript.
Comment 7: The discussion is comprehensive and aligns with the results. However, the conclusion section could benefit from a statement outlining perspectives for future research.
Response 7: Thank you for your positive feedback and insightful suggestion. Following your advice, we have revised the conclusion section to include perspectives for future research (Ln 382–391).
Comment 8: Minor typographical errors were found throughout the manuscript. These should be addressed during revision.
Response 8: Thank you for pointing out the typographical errors. We have carefully reviewed the manuscript and corrected all identified errors.
Comment 9: In-text citations do not consistently follow the journal's formatting guidelines. This should be corrected.
Response 9: Thank you for highlighting the inconsistencies in in-text citations. We have thoroughly reviewed the manuscript and ensured that all citations now adhere to the journal’s formatting guidelines. Necessary corrections have been made to maintain consistency throughout.
Comment 10: Scientific names must be italicized where applicable (e.g., Ln 218, 421, 426).
Response 10: Thank you for your observation regarding the formatting of scientific names. We have carefully reviewed the manuscript and ensured that all scientific names are italicized where applicable.
Comment 11: The format of article titles in the reference list is inconsistent. Some titles begin with uppercase letters, while others begin with lowercase letters. All references should follow the journal's uniform formatting guidelines (e.g., Ln 362, 385, 407).
Response 11: Thank you for your valuable suggestion. We have made the necessary changes to ensure consistency in the reference list formatting.
We hope these revisions adequately address your comments and enhance the overall quality of our manuscript. Thank you once again for your invaluable feedback.

Reviewer 2 Report
Comments and Suggestions for Authors
Please check the attachment.

Many thanks for invitation to review the manuscript!
Author Response
Thank you for your positive feedback on the manuscript and recognition of our research's contribution. We are deeply grateful for your constructive comments. We have highlighted the revisions using the track changes mode and carefully addressed the issues raised to improve the clarity and quality of the text. The necessary adjustments have been made, and we believe these changes will further enhance the manuscript. Below, we will respond to each of your points in detail:
Comment 1: Too much common knowledge in the part of Introduction. Paragraph 1 and 2 should be merged and reorganized to briefly describe the damage of O. aegyptiaca and biological control is an effective way to solve problem. The effect mechanism of root microbiome on plant should be deeply analyzed especially at genomic level.
Response 1: Thank you for your constructive comments on the Introduction. We agree that the first two paragraphs contain some common knowledge, and we have revised this section to merge and reorganize the content for greater conciseness (Ln 59-68).
Comment 2: Results, the analysis seems not enough. Please deeply investigate the biological implication of the results.
Response 2: We gratefully appreciate for your valuable suggestion. We have made the changes as suggested (Ln 222-224, Ln 243-245, Ln 262-265 and Ln 285-287).
Comment 3: Figure 1, the orders of materials in A and B are not same as C.
Response 3: We gratefully appreciate for your valuable suggestion. We have made the changes as suggested
Comment 4: Figure 1, the ambiguity in the illustration of Figure 1 (C), please clearly indicate what traits impact on such as the root morphology or others?
Response 4: We gratefully appreciate for your valuable suggestion. We have marked the parasitic Orobanche aegyptiaca with arrows in the Figure 1 (C).(Ln 196).
Comment 5: P6, Line 214-215, the sentence is not professional English expression.
Response 5: We gratefully appreciate for your valuable suggestion. We have made the changes as suggested (Ln 207-208).
Comment 6: Figure 2, The ‘Difference’ in the title should be impact due to the PCA analysis mainly used to analyze the impact factors.
Response 6: We gratefully appreciate for your valuable suggestion. We have made the changes as suggested.
Comment 7: Figure 3, please clearly indicate what are the left and right in the A and B figures in the illustration. Generally, one illustration for one image.
Response 7: We gratefully appreciate for your valuable suggestion. We have made the changes as suggested (Ln 246-248).
Comment 8: Figure 5, the water of Fig.5E seems different to others. Is there water stress on the germination of J1 treat?
Response 8: We gratefully appreciate for your valuable suggestion. The germination of J1 treatment is not subject to water stress.
Comment 9: The title of Figure has no logic relationship with the title of 3.4, please change the description to echo each other.
Response 9: We appreciate for your valuable comment. We have made the changes as suggested (Ln 274-279).
Comment 10: The discussion is not fully discussed and focus on how to recruit beneficial microbes and the possible effect mechanism.
Response 10: We appreciate for your valuable comment. We have made the changes as suggested (Ln 361-370).

Reviewer 3 Report
Comments and Suggestions for Authors
Dear Authors,
The manuscript is well-written and presents valuable insights. To enhance its overall quality and ensure clarity for the readers, I suggest a few minor modifications. Addressing these points will further strengthen the manuscript's impact and comprehensibility.
Comments on the Materials and Methods and results sections:
Line 84–85:
1. Could you provide additional details about the climatic conditions during the study period, such as temperature, precipitation, and other relevant environmental factors? This information would help contextualize the study's findings.
2. Was there any specific crop rotation or soil preparation method used in the field prior to seed collection? This is important to understand any potential impacts on O. aegyptiaca seed production or viability.
3. Were the seeds collected from multiple plants or a single plant? If from multiple, please explain how the plants were selected and whether any criteria or randomization was applied to ensure representative sampling.
4. How were the seeds processed and stored after collection to ensure viability? Details on drying, storage temperature, and duration would be valuable.
Line 101–102 and in line 177 J1 metabolites impact on O. aegyptiaca parasitism and plant growth
1. How were O. aegyptiaca infestation levels recorded? Were they assessed quantitatively (e.g., counting attachments per plant) or qualitatively (e.g., using categories like light, moderate, severe)? Providing this information would clarify the methodology.
2. Were the infestation levels consistent across all tomato plants, or was there notable variability? If variability existed, how was it accounted for in the analysis?
3. How was plant height measured? For instance, was it measured from the base to the top of the highest leaf or stem? Please specify whether the measurements were taken manually or using a particular tool.
Comments on Figures:
For all figures, I recommend revising the legends to explain the meaning of each initial or abbreviation in the legend, so readers can understand the figure without referring to the main text. Clearly describe all axes, symbols, markers, and any statistical significance indicators (e.g., asterisks for p-values). Ensure the legends are self-explanatory, allowing readers to interpret the data effectively.
Author Response
Thank you for your positive feedback and constructive suggestions. We appreciate your recognition of the manuscript's value and have carefully considered your recommendations. We have already used the track changes mode to highlight these modifications, in order to enhance the clarity and overall quality of the manuscript. These changes aim to improve readability and ensure that the key findings are effectively communicated to the readers. We believe these revisions will contribute to a stronger and more impactful manuscript. Below, we address each of your points:
Comment 1: Line 84-85: Could you provide additional details about the climatic conditions during the study period, such as temperature, precipitation, and other relevant environmental factors? This information would help contextualize the study's findings.
Response 1: We gratefully appreciate for your valuable suggestion. We have made the changes as suggested (Ln 99-102).
Comment 2: Was there any specific crop rotation or soil preparation method used in the field prior to seed collection? This is important to understand any potential impacts on O. aegyptiaca seed production or viability.
Response 2: We gratefully appreciate for your valuable suggestion. Before seed collection, there were no specific crop rotation or soil preparation methods. The seeds were collected from fields that had been planted with processing tomatoes for many years.
Comment 3: Were the seeds collected from multiple plants or a single plant? If from multiple, please explain how the plants were selected and whether any criteria or randomization was applied to ensure representative sampling.
Response 3: We gratefully appreciate for your valuable suggestion. We have made the changes as suggested (Ln 81-83).
Comment 4: How were the seeds processed and stored after collection to ensure viability? Details on drying, storage temperature, and duration would be valuable.
Response 4: We gratefully appreciate for your valuable suggestion. We have made the changes as suggested (Ln 83-84).
Comment 5: Line 101-102 and in line 177 J1 metabolites impact on O. aegyptiaca parasitism and plant growth. How were O. aegyptiaca infestation levels recorded? Were they assessed quantitatively (e.g., counting attachments per plant) or qualitatively (e.g., using categories like light, moderate, severe)? Providing this information would clarify the methodology.
Response 5: We sincerely appreciate the valuable comments. We have checked the literature carefully and added references to the assessed quantitatively in the revised manuscript. (Ln 106-107).
Comment 6: Were the infestation levels consistent across all tomato plants, or was there notable variability? If variability existed, how was it accounted for in the analysis?
Response 6: We sincerely appreciate the valuable comments. There are significant differences in the parasitism levels among all tomato plants, as indicated in Figure 1.(Ln 184-196).
Comment 7: How was plant height measured? For instance, was it measured from the base to the top of the highest leaf or stem? Please specify whether the measurements were taken manually or using a particular tool.
Response 7: We sincerely appreciate your valuable comments. The plant height was manually measured by positioning one end of the ruler at the base of the plant (typically where the soil surface meets the main stem) and then extending it up to the tip of the tallest stem on the plant.
Comment 8: Comments on Figures: For all figures, I recommend revising the legends to explain the meaning of each initial or abbreviation in the legend, so readers can understand the figure without referring to the main text. Clearly describe all axes, symbols, markers, and any statistical significance indicators (e.g., asterisks for p-values). Ensure the legends are self-explanatory, allowing readers to interpret the data effectively.
Response 8: Thank you so much for your careful check and these problems has been corrected in the revised manuscript, we feel sorry for our carelessness.
